# IMPA2 Downregulation Enhances mTORC1 Activity and Restrains Autophagy Initiation in Metastatic Clear Cell Renal Cell Carcinoma

**DOI:** 10.3390/jcm9040956

**Published:** 2020-03-30

**Authors:** Chia-Hao Kuei, Hui-Yu Lin, Hsun-Hua Lee, Che-Hsuan Lin, Jing-Quan Zheng, Kuan-Chou Chen, Yuan-Feng Lin

**Affiliations:** 1Graduate Institute of Clinical Medicine, College of Medicine, Taipei Medical University, Taipei 11031, Taiwan; pplay1028@gmail.com (C.-H.K.); candycarol0227@gmail.com (H.-Y.L.); kaorulei@yahoo.com.tw (H.-H.L.); 16044@s.tmu.edu.tw (J.-Q.Z.); kuanchou@tmu.edu.tw (K.-C.C.); 2Department of Urology, Division of Surgery, Cardinal Tien Hospital, Xindian district, New Taipei City 23148, Taiwan; 3Department of Breast Surgery and General Surgery, Division of Surgery, Cardinal Tien Hospital, Xindian district, New Taipei City 23148, Taiwan; 4Department of Neurology, Shuang Ho Hospital, Taipei Medical University, New Taipei City 23561, Taiwan; 5Department of Neurology, School of Medicine, College of Medicine, Taipei Medical University, Taipei 11031, Taiwan; 6Department of Neurology, Vertigo and Balance Impairment Center, Shuang Ho Hospital, Taipei Medical University, New Taipei City 23561, Taiwan; 7Graduate Institute of Medical Sciences, College of Medicine, Taipei Medical University, Taipei 11031, Taiwan; cloudfrank@gmail.com; 8Department of Otolaryngology, Taipei Medical University Hospital, Taipei Medical University, Taipei 11031, Taiwan; 9Department of Critical Care Medicine, Shuang Ho Hospital, Taipei Medical University, New Taipei City 23561, Taiwan; 10Department of Urology, School of Medicine, College of Medicine, Taipei Medical University, Taipei 11031, Taiwan; 11Department of Urology, Shuang Ho Hospital, Taipei Medical University, New Taipei City 23148, Taiwan; 12Cell Physiology and Molecular Image Research Center, Wan Fang Hospital, Taipei Medical University, Taipei 11696, Taiwan

**Keywords:** renal cell carcinoma, IMPA2, mTOR, autophagy, metastasis

## Abstract

Although mTOR inhibitors have been approved as first-line therapy for treating metastatic clear cell renal cell carcinoma (ccRCC), the lack of useful markers reduces their therapeutic effectiveness. The objective of this study was to estimate if inositol monophosphatase 2 (IMPA2) downregulation refers to a favorable outcome in metastatic ccRCC receiving mTOR inhibitor treatment. Gene set enrichment analysis predicted a significant activation of mTORC1 in the metastatic ccRCC with IMPA2 downregulation. Transcriptional profiling of IMPA2 and mTORC1-related gene set revealed significantly inverse correlation in ccRCC tissues. Whereas the enforced expression of exogenous IMPA2 inhibited the phosphorylation of Akt/mTORC1, artificially silencing IMPA2 led to increased phosphorylation of Akt/mTORC1 in ccRCC cells. The pharmaceutical inhibition of mTORC1 activity by rapamycin reinforced autophagy initiation but suppressed the cellular migration and lung metastatic abilities of IMPA2-silenced ccRCC cells. In contrast, blocking autophagosome formation with 3-methyladenine rescued the mitigated metastatic potential in vitro and in vivo in IMPA2-overexpressing ccRCC cells. Our findings indicated that IMPA2 downregulation negatively activates mTORC1 activity and could be a biomarker for guiding the use of mTOR inhibitors or autophagy inducers to combat metastatic ccRCC in the clinic.

## 1. Introduction

Renal cell carcinoma (RCC) denotes cancer originating from the renal epithelium and accounts for >90% of cancers in the kidney. The major subtypes of RCC are clear cell RCC (ccRCC), papillary RCC (pRCC) and chromophobe RCC (chRCC) [1]. ccRCC is the most common subtype (approximately 75%) and accounts for the majority of deaths related to kidney cancer [2]. Despite nephrectomy with curative intent, approximately 30% of ccRCC patients with a localized disease eventually develop metastases, which are associated with high mortality [3]. Multitargeted receptor tyrosine kinase inhibitors pazopanib and sunitinib have been developed as first-line therapies for metastatic ccRCC via inhibiting the activity of vascular endothelial growth factor and mechanistic target of rapamycin (mTOR), also referred to as the mammalian target of rapamycin and belonging to a member of the phosphatidylinositol 3-kinase (PI3K)-related protein kinases, but the treatment response is varied, and most cancers eventually progress [4]. Therefore, the increased genomic and molecular understanding of metastatic ccRCC may provide a method for the future design of personalized clinical management plans.

Inositol monophosphatase (IMPA) has two subtypes, IMPA1 and IMPA2, which catalyze the hydrolysis of inositol monophosphate (IP1) into free inositol, which is required for the phosphoinositol signaling pathway [5]. Lithium, an effective therapy for manic depression, has been shown to inhibit IMPase activity at clinically relevant concentrations [6,7]. A recent report demonstrated that lithium directly inhibits the enzyme activity of IMPA1, not IMPA2, in the cell-based assays [8]. Additionally, biochemical studies comparing the enzyme activity of IMPA2 to that of IMPA1 suggest that IMPA2 has an in vivo function separate from that of IMPA1 [9]. While the molecular effects of lithium are partially mimicked by IMPA1 knockout mice in a brain region-dependent manner [10], IMPA2 knockout mice lack lithium-like behavioral and molecular effects [11]. In contrast, male transgenic mice with IMPA2 knockout exhibited a lithium-resistant phenotype [12]. These findings demonstrated that IMPA1 and IMPA2 exhibit a different response to lithium treatment and have distinct biological functions. Although, to date, there are limited reports on the role of IMPase in cancer development and metastatic progression, we recently reported that the posttranscriptional repression of IMPA2, but not IMPA1, promotes metastatic progression in ccRCC [13]. Nevertheless, further experiments are still needed to explore the comprehensive mechanism by which IMPA2 downregulation drives ccRCC metastasis.

A previous report revealed that lithium induces autophagy which is a spherical structure with double layer membranes and involved in the intracellular degradation system for abnormal proteins or damaged organelles independent of the activity of mTOR, a negative regulator of autophagy, by inhibiting inositol monophosphatase in neural cells [14]. In colorectal cancer, lithium treatment was shown to enhance the therapeutic responsiveness to chemotherapeutic agents by suppressing the activity of PI3K/Akt, an upstream regulator of mTOR, by triggering autophagy initiation [15,16]. Since previous reports suggested that IMPA2 might not be the target by which lithium elicits its effects, we investigated the activity of mTOR and autophagy in metastatic ccRCC with IMPA2 downregulation. In this study, by using in silico gene set enrichment analysis (GSEA) and cell-based assays, we found that metastatic ccRCC with IMPA2 downregulation exhibits increased Akt/mTORC1 activity and restrained autophagy initiation. Pharmaceutical inhibition of mTORC1 with rapamycin restored autophagy initiation and ultimately suppressed the metastatic potential of IMPA2-silenced ccRCC cells in vitro and in vivo. These findings suggest that IMPA2 could be a biomarker for predicting the therapeutic effectiveness of mTOR inhibitors in combating metastatic ccRCC in the clinic.

## 2. Materials and Methods

### 2.1. Clinical and Molecular Data for RCC Patients

The clinical data, including age, sex, cancer grade, cancer stage, Tumor, Node and Metastasis (TNM) stage, and overall survival (OS) time for The Cancer Genome Atlas (TCGA) ccRCC patients, were collected from the University of California, Santa Cruz (UCSC) Xena website (http://xena.ucsc.edu/welcome-to-ucsc-xena/). The molecular data obtained by RNAseq (polyA þ Illumina HiSeq) analysis of the TCGA ccRCC cohort were also downloaded from the UCSC Xena website. For gene set enrichment analysis (GSEA), the transcriptional profile of all somatic genes in the ccRCC tissues derived from patients who were stratified as a low-level IMPA2 expression in the Kaplan–Meier analysis of our previous report [13] and recorded to be dead in the overall survival condition and positive for pathologic M stage and who were stratified as a high-level IMPA2 expression in the Kaplan–Meier analysis of our previous report [13] and recorded to be alive in the overall survival condition and negative for pathologic M stage were subjected to Pearson’s correlation tests against IMPA2 expression. The obtained results of Pearson correlation coefficient (r) for the IMPA2 co-expression in the tested tissues from the two grouped ccRCC patients were then analyzed by GSEA program using hallmarks gene sets deposited in the Molecular Signatures Database (https://www.gsea-msigdb.org/gsea/msigdb).

### 2.2. Cell Culture

The human renal adenocarcinoma cell lines ACHN and A498 were purchased from American Type Culture Collection (ATCC); the cells were maintained in minimal essential medium (MEM) supplemented with 10% FBS, penicillin (100 U/mL) and streptomycin (100 μg/mL). The human renal adenocarcinoma cell lines Caki-1 and Caki-2 were purchased from ATCC; the cells were maintained in McCoy’s 5A medium supplemented with 10% FBS, penicillin (100 U/mL) and streptomycin (100 μg/mL). All cells were incubated in 5% CO_2_ at 37 °C. All media and supplements were purchased from Gibco Life Technologies. 293T cells were cultured in DMEM with 10% FBS and incubated at 37 °C with 5% CO_2_. All cell lines were routinely authenticated on the basis of short tandem repeat (STR) analysis, morphologic and growth characteristics and mycoplasma detection.

### 2.3. Cellular Migration Assays

In vitro cellular migration ability was assessed by using the Boyden Chamber Assay (NeuroProbe, Gaithersburg, MD, USA). Cells (1.5 × 10^4^) in serum-free culture medium were added to the upper chamber of the device on a polyvinylidene difluoride (PVDF) membrane with an 8.0 μm pore and precoated with 10 μg/mL fibronectin (Invitrogen) at the opposite site, and the lower chamber was filled with 10% FBS culture medium. After a 3 h incubation, the membrane was soaked in methanol for 10 min and stained with Giemsa, which was diluted 10-fold by double-distilled water, for 1 hour. Then, the membrane was attached to slides, and the cells on the upper side of the membrane were carefully removed with a cotton swab. The cells on the lower side were photographed. The migrated cells were quantified by counting the cells in three random areas under a microscope at 400× magnification.

### 2.4. Lentivirus Production and Transduction

Nonsilencing control and IMPA2 shRNAs were purchased from the RNAi Core of Academia Sinica (Taipei, Taiwan). The sequences of the IMPA2 shRNAs used are as follows: shRNA1, 5’-GCTGTTCGACAAGAGCTTGAA-3’; and shRNA2, 5’-AGAGGGAGTTGTCACGCTACA-3’. The coding sequence of the IMPA2 gene was amplified from human cDNA using a standard polymerase chain reaction (PCR) protocol with paired primers (forward: 5’-TAAGCAGCTAGCATGAAGCCGAGCGGC-3’; reverse: 5’-TGTTTAGAATTCTCACTTCTCATCATCCCGC-3’) and then subcloned into the pLAS3w/Pbsd vector with NheI/EcoRI restriction sites. Lentiviruses were produced by cotransfecting 293T cells with shRNA or IMPA2-expressing plasmids, envelope plasmids (pMD.G) and packaging plasmids (pCMV-dR8.91). The 293T cells were incubated for 24 h, and then the culture medium was removed and refreshed. The viral supernatants were harvested, and the titers were determined posttransfection for 48 h. Green fluorescent protein (GFP) and GFP-LC3 fusion-expressing lentiviral particles were purchased from Cell Biolabs (San Diego, CA, USA). RCC cells were transduced with the lentiviruses at a multiplicity of infection of 10 in the presence of polybrene and selected using puromycin after infection to generate a stable clone.

### 2.5. Western Blotting Assay

Aliquots of 100 μg of total protein and HR Pre-Stained Protein Marker 10-170 kDa (Biotools, New Taipei City, Taiwan) were loaded into each well of an SDS gel, separated by electrophoresis and then transferred to PVDF membranes. The membranes were incubated with blocking buffer (5% skim milk in TBS containing 0.1% Tween-20) for 2 h at room temperature. The samples were incubated with primary antibodies against phosphorylated Akt (p-Akt), Akt, p-mTOR, mTOR and LC3I/II (Cell Signaling, Danvers, MA, USA), as well as GAPDH (AbFrontier, Seoul, Korea) overnight at 4 °C. After extensive washing, the membranes were incubated with a peroxidase-labeled secondary antibody for 1 h at room temperature. Immunoreactive bands were visualized by using an enhanced chemiluminescence system (Amersham Biosciences, Tokyo, Japan).

### 2.6. Animal Experiments

NOD/SCID mice were obtained from the National Laboratory Animal Center in Taiwan and maintained in compliance with institutional policy. All animal procedures were approved by the Institutional Animal Care and Use Committee at Taipei Medical University. For the in vivo lung metastatic colonization assay, cell suspensions (1 × 10^5^ cells in 100 μL PBS) derived from A498 cells without or with IMPA2 knockdown and ACHN cells without or with IMPA2 overexpression were implanted into the mice through tail vein injection. The mice were sacrificed, and the lungs were obtained for histological analysis. Metastatic lung nodules were quantified after staining with H&E using a dissecting microscope.

### 2.7. Statistical Analyses

Statistical analyses were performed using SPSS 17.0 software (Informer Technologies, Roseau, Dominica). Student’s *t*-tests were utilized to compare the expression levels of mTORC1 and genes from the autophagy gene sets that were obtained from the Molecular Signatures Database (MSigDB, http://software.broadinstitute. org/gsea/msigdb) in TCGA ccRCC cohorts. Pearson’s test was performed to estimate the correlations among the expression of IMPA2 and the mTORC1 autophagy gene sets and to evaluate the coexpression of IMPA2 with somatic genes in the TCGA ccRCC cohort. Survival probabilities were determined by Kaplan–Meier analysis and log-rank tests. A nonparametric Friedman test was used to analyze data from 3 or more related samples. *p* values <0.05 in all analyses were considered statistically significant.

## 3. Results

### 3.1. IMPA2 Downregulation Accompanied by Enhanced mTORC1 Activity Correlates with Metastatic Progression and Poor Prognosis in ccRCC Patients

Previously, we demonstrated that low-level IMPA2 expression is associated with high risk for cancer metastasis and poor prognosis in TCGA ccRCC patients [13]. According to the stratification using IMPA2 levels in a Kaplan–Meier analysis against TCGA ccRCC patients in our previous report, here we performed Pearson’s test to examine the coexpression of the IMPA2 transcript with other somatic genes in either metastatic ccRCC with low-level IMPA2 or nonmetastatic ccRCC with high-level IMPA2 to ascertain the possible mechanism by which IMPA2 downregulation promotes ccRCC metastasis (Figure 1A). The obtained results of Pearson’s correlation test were further used to perform an in silico gene set enrichment analysis (GSEA, https://www.gsea-msigdb.org/gsea) (Figure 1A). The computational simulation by GSEA software demonstrated that the expression of the mTORC1 gene set, which putatively reflects the status of mTORC activity, inversely correlates with the IMPA2 levels in metastatic or nonmetastatic ccRCC (Figure 1B). Moreover, the expression of the mTORC1 gene set in metastatic ccRCC was higher than that in nonmetastatic ccRCC from the TCGA ccRCC cohort (Figure 1C). Pearson’s correlation test also showed that the expression of IMPA2 and the mTORC1 gene set in the TCGA ccRCC cohort, regardless of pathologic M status, appears to be negatively correlated (r = −0.382) with statistical significance (*p* = 7.92 × 10^−19^) (Figure 1D). Kaplan–Meier analysis demonstrated that high mTORC1 gene set levels are associated with poor overall survival probability in the TCGA ccRCC cohort (Figure 1E). Notably, another Kaplan–Meier analysis revealed that the signature that combined low-level IMPA2 and high-level mTORC1 gene set expression predicts poor prognosis in TCGA ccRCC patients (Figure 1F). Furthermore, the Cox regression analysis indicated that low-level IMPA2 expression and the signature that combines low-level IMPA2 and high-level mTORC1 gene set expression, but not high-level mTORC1 gene set expression alone, act as independent risk factors in the multivariate analysis, even though all of them appear to be poor prognostic markers for predicting an unfavorable outcome in the univariate analysis of TCGA ccRCC patients in the univariate analysis (Appendix A). In addition, the Chi-square test showed that the signature that combines low-level IMPA2 and high-level mTORC1 gene set expression is extensively detected in ccRCC derived from male patients who are classified as having higher pathologic T status (T3 and T4), pathologic M1, higher pathologic stage (III and IV) or higher neoplasm grade (G3 and G4) (Appendix A). These findings suggest that IMPA2 downregulation might restore mTORC1 activity to promote cancer progression, e.g., metastasis, in ccRCC.

### 3.2. IMPA2 Expression Negatively Regulates the Activation of the Akt/mTORC1 Pathway and Associates with Autophagy Formation in ccRCC Cells

To determine the correlation between IMPA2 expression and mTORC1 activity, we next performed Western blot analysis to examine the phosphorylation status of Akt, an upstream regulator of mTORC1, and mTORC1 proteins in ccRCC cell lines with different cellular migration abilities (Figure 2A). The data demonstrated that ACHN cells with a stronger migration ability express lower IMPA2 protein levels and enhanced phosphorylation of Akt and mTORC1 proteins; A498 cells with a poorer migration ability exhibited higher IMPA2 protein levels but relatively less phosphorylation of Akt and mTORC1 proteins (Figure 2B). Similarly, the transcriptional profiling of IMPA2 and the mTORC1 gene set appeared to be a significantly (*p* = 0.046) inversely correlated in a panel of RCC cell lines deposited in the Cancer Cell Line Encyclopedia database (https://portals.broadinstitute.org/ccle) (Figure 2C). Our previous report demonstrated that IMPA2 knockdown in poorly metastatic A498 cells promotes but IMPA2 overexpression in highly metastatic ACHN cells suppresses cellular migration ability [13]. Here we found that artificially silencing IMPA2 expression enhanced the intracellular levels of phosphorylated Akt and mTORC1 proteins in poorly metastatic A498 cells (Figure 2D). Conversely, the enforced expression of exogenous IMPA2 reduced the phosphorylation levels of Akt and mTORC1 proteins in highly metastatic ACHN cells (Figure 2E).

Since several lines of evidence have shown that the Akt/mTORC1 signaling axis is a negative regulator of autophagy formation, we thus detected the levels of the autophagy assembly initiator Beclin 1 and of LC3 conversion from LC3-I to LC3-II, which is a phosphatidylethanolamine- conjugated form of LC3-I that acts as a critical initiator for autophagy assembly. Our data showed that Beclin 1 and LC3-II levels inversely correlated with cellular migration ability and phosphorylation levels of Akt/mTORC1 proteins in the detected ccRCC cell lines (Figure 3A). Robustly, autophagosome accumulation was dominant in A498 cells but not ACHN cells (Figure 3B). Accordingly, the transcriptional profiling of IMPA2 and the autophagy gene set obtained from hallmark gene sets of Molecular Signatures Database (https://www.gsea-msigdb.org/gsea/msigdb) showed a significant (*p* = 0.0049) positive correlation in RCC cell lines from the Cancer Cell Line Encyclopedia database (Figure 3C). IMPA2 knockdown reduced Beclin 1 expression and LC3-II production in poorly metastatic A498 cells (Figure 3D). In contrast, IMPA2 overexpression enhanced Beclin 1 expression and LC3-II production in highly metastatic ACHN cells (Figure 3E).

To evaluate the determinant role of autophagy induction in IMPA2-associated metastatic progression of ccRCC, we employed rapamycin (Rapa, mTORC1 inhibitor) and 3-methyladenine (3-MA, autophagy inhibitor) to modulate autophagy activity in the detected ccRCC cells. In poorly metastatic A498 cells, IMPA2 knockdown abrogated the production of LC3-II (Figure 4A) but promoted cellular migration ability (Figure 4B,C) and lung metastatic potential (Figure 4D,E). The addition of rapamycin to inhibit mTORC1 activity dose-dependently restored the production of LC3-II (Figure 4A) but diminished the cellular migration ability (Figure 4B,C) and lung metastatic potential (Figure 4D,E) in IMPA2-silenced A498 cells. On the other hand, IMPA2 overexpression promoted the production of LC3-II (Figure 5A) but suppressed the cellular migration (Figure 5B,C) and lung colony formation (Figure 5D,E) abilities in highly metastatic ACHN cells. Dramatically, the pharmaceutical inhibition of autophagy assembly by 3-MA blocked the production of LC3-II (Figure 5A) but rescued the cellular migration (Figure 5B,C) and lung colony formation (Figure 5D,E) abilities in IMPA2-overexpressing ACHN cells. These findings indicate that IMPA2 downregulation promotes the metastatic progression of ccRCC via activating the Akt/mTORC1 pathway to restrain autophagy formation.

### 3.3. IMPA2 Downregulation Combined with Decreased Autophagy is Associated with a Poor Overall Survival Rate in ccRCC Patients

The transcriptional profiling of TCGA ccRCC tissues revealed that the expression levels of autophagy gene set were significantly (*p* = 0.00009) downregulated in metastatic ccRCC compared to nonmetastatic ccRCC (Figure 6A). Moreover, the levels of the autophagy gene set expression appeared to positively correlate with IMPA2 expression in TCGA ccRCC samples (Figure 6B). Regarding overall survival probability, low-level autophagy gene set expression was strongly associated with poor prognosis in the Kaplan –Meier analysis of the TCGA ccRCC cohort (Figure 6C). Importantly, another Kaplan–Meier analysis revealed that the signature that combines low-level IMPA2 and autophagy gene set predicted a poor overall survival rate in TCGA ccRCC patients (Figure 6D). The Cox regression test indicated that low-level autophagy gene set expression and the signature that combines low-level IMPA2 and autophagy gene set expression, but not low-level IMPA2 expression alone, served as independent risk factors in the multivariate analysis, even though all of them appear to be poor prognostic markers for estimating an unfavorable outcome in the univariate analysis of TCGA ccRCC patients (Figure 6E). In addition, the Chi-square test showed that the signature that combines low-level IMPA2 and autophagy gene set expression is extensively detected in ccRCC derived from male patients who are classified as having higher pathologic T status (T3 and T4), pathologic M1, higher pathologic stage (III and IV) or higher neoplasm grade (G3 and G4) (Figure 6F). These findings demonstrate that the downregulation of IMPA2 leads to the inhibition of autophagy initiation, owing to enhanced activity of mTORC1, and eventually promotes the metastatic progression of ccRCC.

## 4. Discussion

In this study, the in silico experiments demonstrated that IMPA2 downregulation may correlate with an increased activity of mTORC1 in metastatic ccRCC, and this scenario closely associates with a poorer prognosis in ccRCC patients. IMPA2 knockdown in poorly metastatic A498 cells enhanced, but IMPA2 overexpression in highly metastatic ACHN cells suppressed the activation of Akt/mTORC1, a negative regulator for autophagy initiation, and the cellular metastatic potentials in vitro and in vivo. The pharmaceutical inhibition of mTORC1 by rapamycin restored autophagy initiation but mitigated the cellular migration ability and lung colony-forming ability of IMPA2-silenced A498 cells. These findings suggest that mTORC1 inhibitors might be useful for treating metastatic ccRCC with IMPA2 downregulation (Figure 6G).

A recent report demonstrated that in 20% of patients, ccRCC is frequently diagnosed at the metastatic stage, and in 30% of the remaining patients, metastases will be detected during follow-up. The mortality rate of patients with metastatic ccRCC is 40% at 5 years [17]. However, over the last decade, the treatment of metastatic ccRCC has seen substantial progress. Nonspecific immunotherapy with high-dose interleukin-2 was initially considered as a standard therapy for metastatic ccRCC. Moreover, the development of therapies targeted against the tyrosine kinase activity of vascular endothelial growth factor (VEGF) and novel immune checkpoint inhibitors significantly increased the overall survival of patients with metastatic ccRCC. Based on evidence from randomized phase III clinical trials, the VEGF tyrosine kinase inhibitors (VEGF-TKIs) sunitinib and pazopanib are the most effective first-line options, especially in favorable and intermediate risk patients [18]. On the other hand, the combination of dual checkpoint inhibitors nivolumab and ipilimumab seems to be the preferred first-line therapy in poor-risk patients [19,20]. In addition to targeting VEGF and immune checkpoints, mTOR was also considered as a target for combating metastatic RCC [21]. Furthermore, in 2007, the United States Food and Drug Administration and the European Medicines Agency approved mTOR inhibitor temsirolimus as a first-line therapy for poor-risk patients with metastatic ccRCC, and the National Comprehensive Cancer Network (NCCN) Kidney Cancer Panel has listed temsirolimus as category 1 for front-line treatment of poor-risk patients since temsirolimus, in the phase III NCT0065468 trial which included 626 patients (69% with poor- and 31% with intermediate-risk characteristics), achieved longer progression-free and overall survival [22]. Nevertheless, patient selection to increase response to specific agents remains a major challenge, and better biomarkers and predictive models are urgently needed. Here, we show that IMPA2 downregulation is concurrent with the enhanced activity of the mTORC1 pathway in metastatic ccRCC. This finding may provide a new strategy for establishing personalized therapy by suggesting that metastatic ccRCC patients with IMPA2 deficiency should receive temsirolimus therapy.

It has been shown that the activation of the Akt/mTOR pathway by the long noncoding RNAs OECC [23] and MetaLnc9 [24] and the transmembrane 7 superfamily member 4 [25] promotes cancer metastasis; conversely, the suppression of the Akt/mTOR pathway in the presence of the ferulic acid derivative FXS-3 [26], cardamonin [27] and microRNA-520a-3p [28] inhibits cancer metastatic potential. Moreover, the association between the activity of the Akt/mTOR pathway and metastatic progression has been reported in various cancer types, including colorectal cancer [29], hepatocellular carcinoma [30], endometrial cancer [31], ovarian cancer [32], gastric cancer [33], melanoma [34], glioma [35], pancreatic ductal adenocarcinoma [36], nasopharyngeal carcinoma [37], osteosarcoma [38], breast cancer [39] and prostate cancer [40]. In renal cell carcinoma, treatment with simvastatin, an inhibitor of HMG-CoA reductase, and bufalin suppressed the metastatic activity of RCC cells that was accompanied by inhibition of the Akt/mTOR pathway [41,42]. Here, we further explored whether RCC cells with high-level IMPA2 expression displayed poor Akt/mTOR activity, as judged by their phosphorylation level. This finding differs from a previous report that IMPase is capable of enhancing PI3K activity to promote the activation of Akt/mTOR. Therefore, further experiments are needed to explain how IMPA2 negatively regulates Akt/mTOR in ccRCC cells.

The role of autophagy during cancer metastasis is still controversial. A recent review article indicated that autophagy is upregulated during cancer metastasis [43]. In contrast, several lines of evidence have illustrated that autophagy is suppressed during the metastatic progression of some cancer types [44,45]. In RCC cells, the induction of autophagy has been implicated to promote epithelial-mesenchymal transition and cellular invasion ability in RCC cells [46]; however, the opposite views were also reported in other studies [47,48]. In this study, we reported that restrained autophagy initiation is detected in ccRCC cells with IMPA2 downregulation and metastatic ccRCC tissues expressing low-level IMPA2 transcripts. Therefore, further studies are needed to elucidate whether the discrepancy is associated with IMPA2 expression in metastatic RCC.

## 5. Conclusions

In summary, our results demonstrated that IMPA2 downregulation promotes the metastatic progression of ccRCC via enhancing the activation of the Akt/mTORC1 pathway. Since the mTOR inhibitor temsirolimus has been approved as a first-line medication for metastatic ccRCC, IMPA2 downregulation may serve as a good biomarker for guiding the use of temsirolimus in the clinical treatment metastatic ccRCC. On the other hand, several autophagy inducers have been developed and employed for combating various malignancies in preclinical studies [49]. IMPA2 downregulation might also be an indicator of the response to autophagy inducers in the future clinical treatment of metastatic ccRCC.

## Figures and Tables

**Figure 1 jcm-09-00956-f001:**
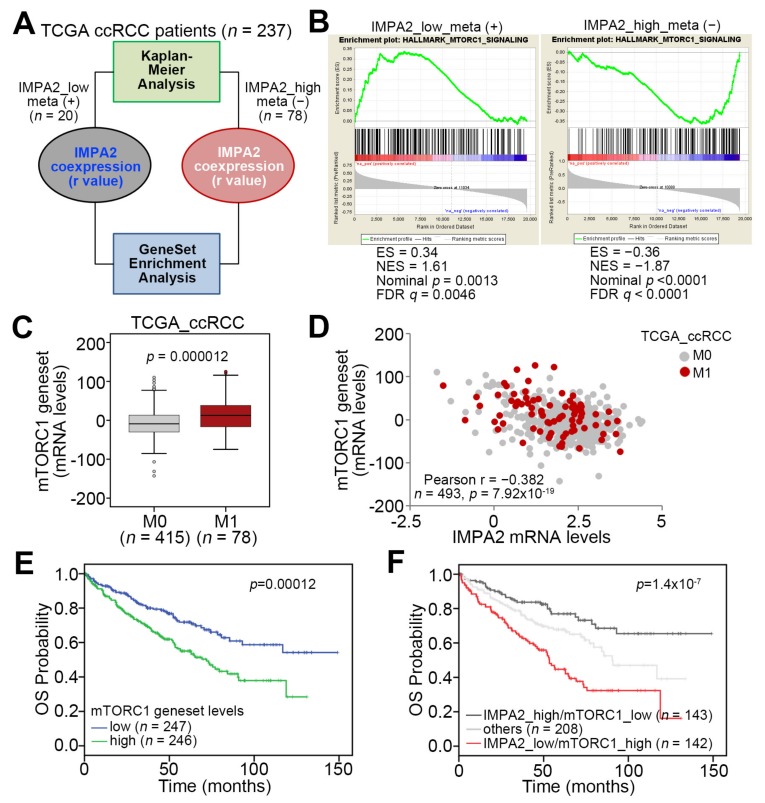
Inositol monophosphatase 2 (IMPA2) downregulation probably correlates with an increased activity of the mTORC1-related pathway and the metastatic progression of clear cell renal cell carcinoma (ccRCC). (**A**) Flowchart for the generation of Pearson correlation coefficient (r) values for the IMPA2 co-expression with somatic genes in the ccRCC tissues derived from the two grouped The Cancer Genome Atlas (TCGA) ccRCC patients in order to perform the computational simulation by gene set enrichment analysis (GSEA) program. The features of two grouped TCGA ccRCC patients are shown in Materials and Methods. (**B**) GSEA plots of Hallmark_mTOC1_signalnig in the IMPA2 co-expression signatures derived from metastatic ccRCC with low-level IMPA2 or non-metastatic ccRCC with high-level IMPA2. (**C**) Boxplot for the mRNA levels of mTORC1 gene set in TCGA ccRCC with pathologic M0 or M1 stage. The band inside the box is the second quartile (the median). The upper and lower lines of the box are the third and first quartiles, respectively. Box plots have lines extending vertically from the whiskers. indicating minimum and maximum of all of the data. The individual points indicate outliers. The significant difference was analyzed by an independent sample t-test. (**D**) Scatchard plot for IMPA2 and mTORC1 gene set mRNA levels from the TCGA ccRCC database. Pearson’s correlation test was used to determine the statistical significance of IMPA2 and mTORC1 gene set co-expression in TCGA ccRCC with pathologic M0 or M1 stage. (**E** and **F**) Kaplan–Meier analyses for mTORC1 gene set mRNA levels (**E**) or the signatures of combined IMPA2 and mTORC1 gene set mRNA levels (**F**) in TCGA ccRCC patients. Others denote the signatures of IMPA2_high/mTORC1_high and IMPA2_low/mTORC1_low. The high and low expression levels of IMPA2 and mTORC1 gene set were determined by the median of their mRNA levels in primary tumors from the TCGA ccRCC database.

**Figure 2 jcm-09-00956-f002:**
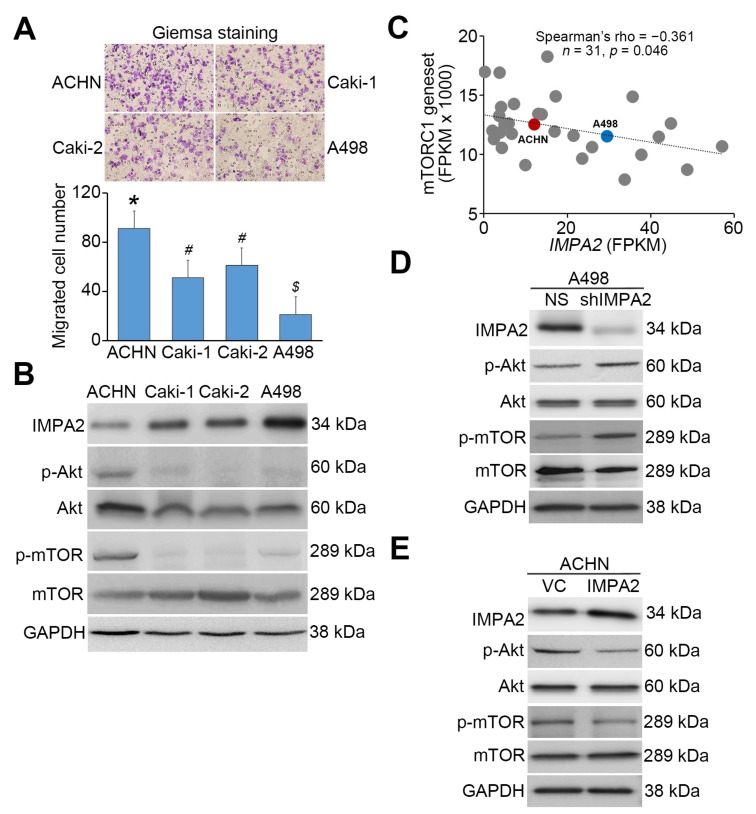
IMPA2 expression negatively regulates Akt/mTORC1 phosphorylation and cellular migration abilities in ccRCC cell lines. (**A**) Giemsa staining (Top) and data from three independent experiments (bottom) for the migrated cells (purple staining) in the 3 h transwell assay for the ccRCC cells. Error bars represent the mean ± SEM of data obtained from three independent experiments and the statistical differences were analyzed by Kruskal Wallis test. Different symbols in each column indicate the statistical significance at *p* < 0.05. (**B**) Western blot analyses for IMPA2, p-Akt (Tyr308), Akt, p-mTOR (Ser2448), mTOR and GAPDH proteins in whole cell lysates derived from the tested ccRCC cell lines ACHN, Caki-1, Caki-2 and A498. (**C**) Correlation of transcriptional profiling between IMPA2 and mTORC1 gene set in a panel of RCC cell lines from Cancer Cell Line Encyclopedia database. Spearman’s correlation test was used to estimate the statistical significance. (**D** and **E**) Western blot analyses for IMPA2, p-Akt, Akt, p-mTOR, mTOR and GAPDH proteins in whole cell lysates derived from A498 cells without (non-silencing, NS) or with IMPA2 knockdown (**D**), and ACHN cells without (vector control, VC) or with IMPA2 overexpression (**E**). In **B**, **D**, **E**, GAPDH was used as an internal control of protein loading.

**Figure 3 jcm-09-00956-f003:**
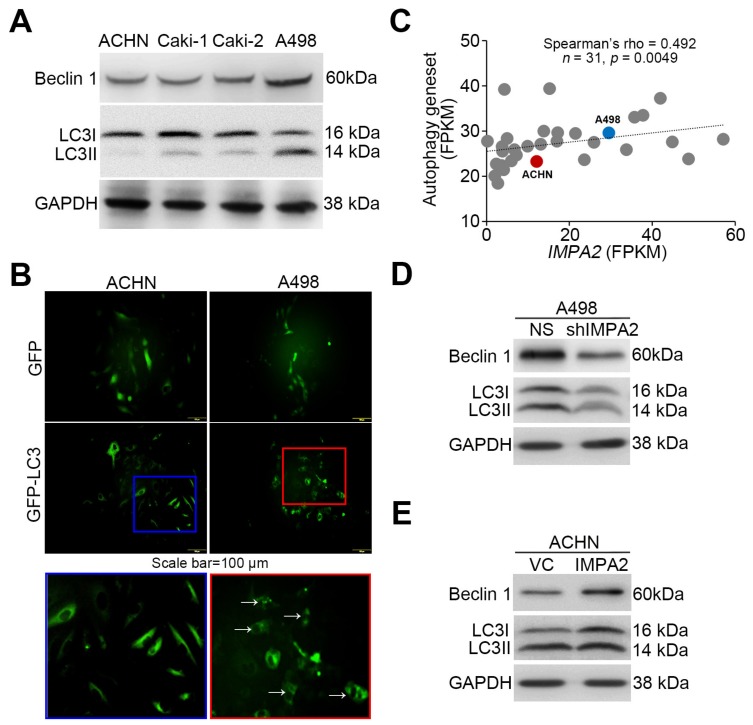
IMPA2 expression modulates autophagy initiation in ccRCC cells. (**A**) Western blot analyses for Beclin 1, LC3I/II and GAPDH proteins in whole cell lysates derived from the tested ccRCC cell lines. (**B**) The fluorescent microscopic observation of ACHN and A498 cells that were stably transfected with gene encoding GFP or GFP-LC3 fusion protein. Two representative pictures captured from GFP-LC3-expressing ACHN (blue region) and A498 (red region) cells were employed to highlight the punctate distribution (white arrows) of autophagosomes in A498, not ACHN, cells. Scale bars in yellow represent 100 μm. (**C**) Correlation of transcriptional profiling between IMPA2 and autophagy gene set in a panel of RCC cell lines from Cancer Cell Line Encyclopedia database. Spearman’s correlation test was used to estimate the statistical significance. (**D** and **E**) Western blot analyses for Beclin 1, LC3I/II and GAPDH proteins in whole cell lysates derived from A498 cells without (non-silencing, NS) or with IMPA2 knockdown (**D**), and ACHN cells without (vector control, VC) or with IMPA2 overexpression (**E**). In **A**, **D**, **E**, GAPDH was used as an internal control of protein loading.

**Figure 4 jcm-09-00956-f004:**
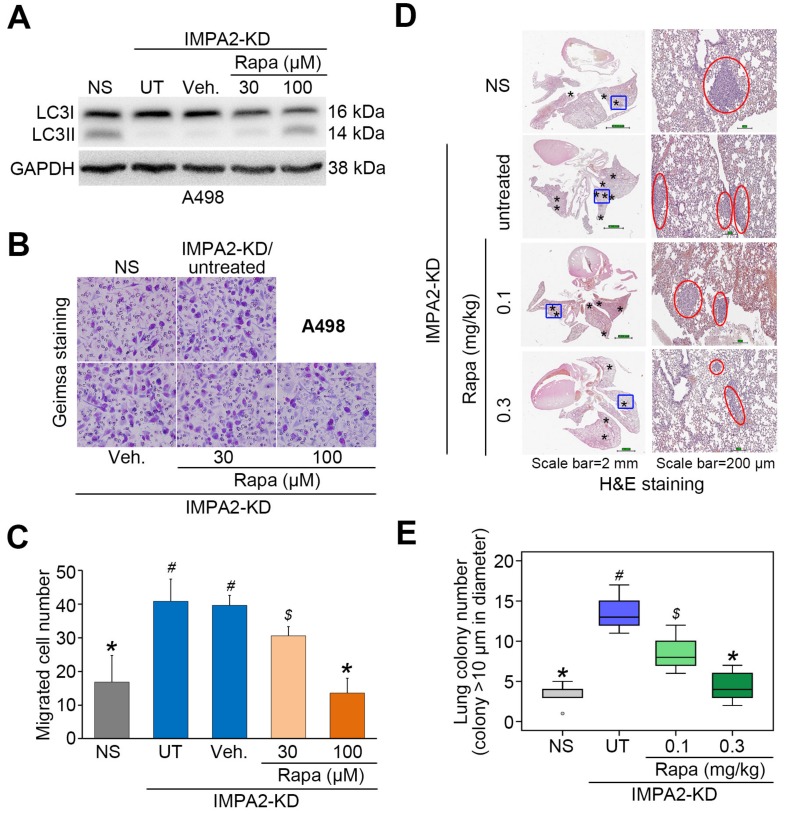
Pharmaceutical inhibition of mTORC1 activity by rapamycin restores autophagy function but compromises the cellular invasion and lung metastatic abilities of IMPA2- silencing A498 cells. (**A**) The results from the Western blot analysis for LC3I/II and GAPDH proteins in whole cell lysates derived from A498 cells without (NS) or with IMPA2 knockdown in the absence (untreated, UT) or presence of solvent control (vehicle, Veh.) or rapamycin (Rapa) at 30 and 100 μM. (**B** and **C**) Giemsa staining (**B**) and cell number (**C**) for the migrated A498 cell variants (dark purple staining) shown in A. Data obtained from three independent experiments are presented as the mean ± SEM. Different symbols *, # and $ indicate the significant differences at *p* < 0.01 analyzed by the nonparametric Friedman test. (**D** and **E**) H&E staining of lung tissues (**D**) and the number of lung tumor colonies (**E**) derived from mice transplanted with the A498 cell variants, shown in A, through tail vein injection for 6 weeks. The symbol “*” indicates the size of tumor colonies over 200 μm in diameter. Tumors are shown in red circles. Statistical significance was analyzed by the nonparametric Kruskal Wallis test. Different symbols *, # and $ in each column indicate the statistical significance at *p* < 0.05.

**Figure 5 jcm-09-00956-f005:**
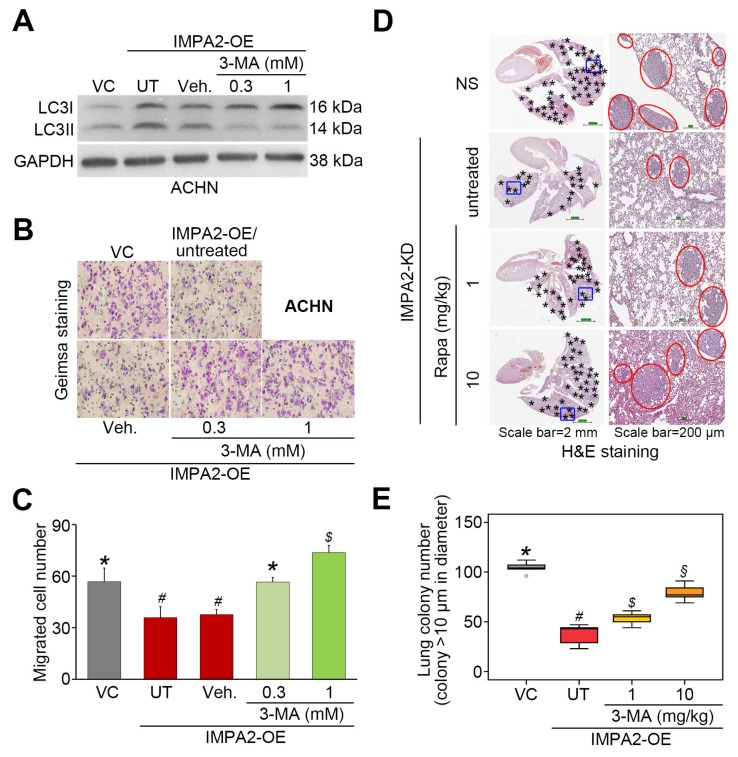
The inhibition of autophagy initiation by 3-MA rescues the metastatic potential of the IMPA2-overexpressing ACHN cells in vitro and in vivo. (**A**) The results from the Western blot analysis for the LC3-I/II and GAPDH proteins derived from ACHN cells without (VC) or with Gαh overexpression (OE) in the absence (untreated, UT) or presence of solvent control (vehicle, Veh.) or the autophagy inhibitor 3-MA (0.3 or 1 mM). (**B** and **C**) Giemsa staining (**B**) and cell number (**C**) of the migrated ACNH cell variants (purple staining) shown in A. Data obtained from three independent experiments are presented as the mean ± SEM. Different symbols *, # and $ indicate the significant differences at *p* < 0.01 analyzed by the nonparametric Friedman test. (**D** and **E**) H&E stained lung tissues (**D**) and the number of lung tumor colonies (**E**) derived from the mice transplanted with ACHN cell variants, shown in A, through tail vein injection for 4 weeks. The symbol “*” indicates the size of tumor colonies over than 200 μm in diameter. Tumors are shown in red circles. Statistical significance was analyzed by the nonparametric Kruskal Wallis test. Different symbols *, #, $ and § in each column indicate the statistical significance at *p* < 0.05.

**Figure 6 jcm-09-00956-f006:**
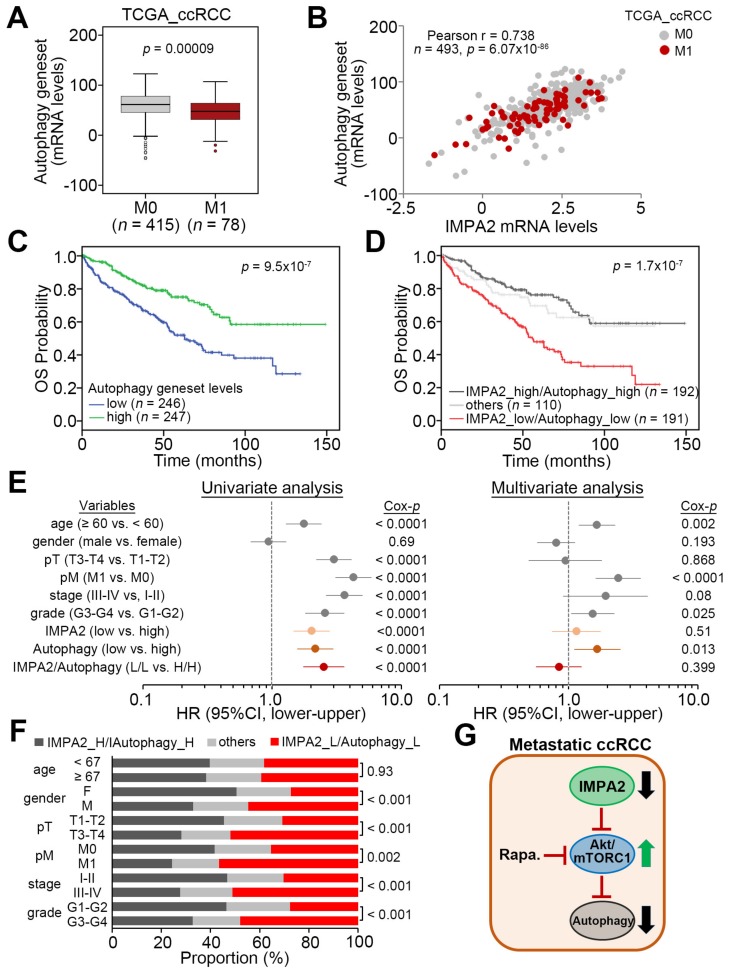
IMPA2 downregulation combined with a reduced autophagy activity correlates with a worse prognosis in ccRCC patients. (**A**) Boxplot for the mRNA levels of autophagy gene set in TCGA ccRCC with pathologic M0 or M1 stage. The band inside the box is the second quartile (the median). The upper and lower lines of the box are the third and first quartiles, respectively. Box plots have lines extending vertically from the whiskers indicating minimum and maximum of all of the data. The individual points indicate outliers. The significant difference was analyzed by an independent sample *t*-test. (**B**) Scatchard plot for IMPA2 and autophagy gene set mRNA levels from the TCGA ccRCC database. Pearson’s correlation test was used to determine the statistical significance of IMPA2 and autophagy gene set co-expression in TCGA ccRCC with pathologic M0 or M1 stage. (**C** and **D**) Kaplan–Meier analyses for autophagy gene set mRNA levels (**C**) or IMPA2 and autophagy gene set mRNA levels (**D**) in TCGA ccRCC patients. The high and low expression levels of IMPA2 and autophagy gene set were determined by the median of their mRNA levels in primary tumors from the TCGA ccRCC database. (**E**) Cox regression test using univariable (left) and multivariable (right) modes for IMPA2, (low vs. high), autophagy gene set (low vs. high) and IMPA2/autophagy gene set (low/low vs. high/high) mRNA levels and other pathological variables including age (median = 60 years, elder vs. younger), gender (male vs. female), pathologic T (pT, T1/T2 vs. T3/T4), pM (M1 vs. M0), stage (III/IV vs. I/II) and grade (III/IV vs. I/II) under the condition of overall survival probability for 493 ccRCC patients from the TCGA database. (**F**) Chi-square test for the combined signature of IMPA2 and autophagy gene set mRNA levels and pathologic variables including age, gender, pT, pM, stage and grade from the TCGA ccRCC database. The high (H) and low (L) expression levels of IMPA2 and autophagy gene set are presented as the signatures IMPA2_H/Autophagy_H, others (IMPA2_H/Autophagy_L and IMPA2_L/Autophagy_H) and IMPA2_L/Autophagy_L. (**G**) Proposed pathway for the IMPA2 downregulation-triggered activation of Akt/mTORC1 and inhibition of autophagy initiation in metastatic ccRCC and a therapeutic value of targeting mTORC1 by its inhibitors, e.g. Rapamycin (Rapa.), to combat metastatic ccRCC with IMPA2 downregulation.

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
