# Peer review of "IMPA2 Downregulation Enhances mTORC1 Activity and Restrains Autophagy Initiation in Metastatic Clear Cell Renal Cell Carcinoma"

_jcm, 2020, doi:10.3390/jcm9040956_

Round 1
Reviewer 1 Report
IMPA2 Downregulation Enhances mTORC1 Activity and Restrains Autophagy Initiation in Metastatic Clear Cell Renal Cell Carcinoma by Kuei et al.
In this manuscript, the authors performed extensive analysis including in silico and experimental analyses. They examined the effect of downregulation of inositol monophosphatase 2 (IMPA2) on enhancing mTORC1 activity and restraining the initiation of autophagy in metastatic clear cell renal cell carcinoma (mccRCC). They presented preliminary evidence that IMPA2 can be a promising biomarker that can guide the treatment of metastatic ccRCC using mTOR inhibitors.
Strengths:
- Identification of a biomarker that will have a great impact on patient management.
- Performing an extensive in silico analysis as well as extensive experimental analysis including both in vitro and in vivo analyses and comparing different cell lines.
Weaknesses:
- Line 146 (1x105 cells in 100 μl PBS were implanted into the mice through tail), please mention what types of cells were used for the experiment.
- Figure 1 A needs to be adjusted and the legend needs more clarification.
- Figure 1 E and F Legends, please explain more.
- Line 218, please add the link to the Cancer Cell Line Encyclopedia database
- Line 222, there is a typo error, (Fig. 2D) to be changed to (Fig. 2E).
- Figure 2 A Legend says (Different letters in each column of b and d indicate the statistical significance at p<0.05) but the bar graph doesn’t show letter d.
- Figure 2 B Legend needs more explanation.
- Line 244, please explain more about the autophagy gene set
- Figure 4 C & E and Figure 5 C & E legends, please indicate in details which letters indicate the significant differences
- Minor grammatical and spell check required for example,
- Line 28 please remove a to read as (Abstract: Although mTOR inhibitors have been approved as first-line therapy for treating)
- Line 34 please change to read as (significant inverse correlation in ccRCC tissues).
- Line 36 please remove an, to read as (artificially silencing IMPA2 led to increased phosphorylation of Akt/mTORC1 in ccRCC cells).
- Line 37 please remove the to read as (The pharmaceutical inhibition of mTORC1 activity by rapamycin reinforced autophagy)
- Line 51 please add a before localized disease.
- Line 150, please remove by to read as (Statistical analyses were performed using SPSS 17.0 software)
- Line 155, remove the extra in
- Line 177, please add are instead of is to read as follows (high mTORC1 gene set levels are associated with poor overall survival probability in the TCGA)
- Line 194, please change performing to perform
- Line 199, please add the before box to read as (The upper and lower lines of the box are the third and first quartiles, respectively).
- Line 280, typo error (please change silecncing to silencing).
- Line 289, please remove than to read as (tumor colonies over 200 µm in diameter)
- Line 290, please add the, to read as (was analyzed by the nonparametric Kruskal Wallis test)
- Line 330, please add the, to read as (The upper and lower lines of the box are the third and first quartiles).
- Line 332, please add an, to read as (The significant difference was analyzed by an independent sample)
Author Response
Reviewer 1:
In this manuscript, the authors performed extensive analysis including in silico and experimental analyses. They examined the effect of downregulation of inositol monophosphatase 2 (IMPA2) on enhancing mTORC1 activity and restraining the initiation of autophagy in metastatic clear cell renal cell carcinoma (mccRCC). They presented preliminary evidence that IMPA2 can be a promising biomarker that can guide the treatment of metastatic ccRCC using mTOR inhibitors.
Strengths:
Identification of a biomarker that will have a great impact on patient management.
Performing an extensive in silico analysis as well as extensive experimental analysis including both in vitro and in vivo analyses and comparing different cell lines.
Response:
Firstly, we should like to thank reviewer #1 very much for the valuable comments and suggestions to our work.
Weaknesses:
Line 146 (1x105 cells in 100 μl PBS were implanted into the mice through tail), please mention what types of cells were used for the experiment.
Response:
As suggested, we have revised the sentence as “cell suspensions (1x105 cells in 100 μl PBS) derived from A498 cells without or with IMPA2 knockdown and ACHN cells without or with IMPA2 overexpression were implanted into the mice through tail vein injection”. Please see lines 158-159 of the revision
Figure 1 A needs to be adjusted and the legend needs more clarification.
Response:
As suggested, we have adjusted Figure 1A as the following figure and revised the figure legend (Please see lines 209-212 of the revision or the follows).
New Figure legend: (A) Flowchart for the generation of Pearson correlation coefficient (r)
values for the IMPA2 co-expression with somatic genes in the ccRCC tissues derived from the two grouped TCGA ccRCC patients in order to perform the computational simulation by GSEA program. The features of two grouped TCGA ccRCC patients are shown in Materials and Methods.
We also provided a detailed information for GSEA experiment in Materials and Methods as the follows (Please see lines 96-105 of the revision).
“For Gene Set Enrichment Analysis (GSEA), the transcriptional profile of all somatic genes in the ccRCC tissues derived from patients who were stratified as a low-level IMPA2 expression in the Kaplan-Meier analysis of our previous report [13] and recorded to be dead in the overall survival condition and positive for pathologic M stage and who were stratified as a high-level IMPA2 expression in the Kaplan-Meier analysis of our previous report [13] and recorded to be alive in the overall survival condition and negative for pathologic M stage were subjected to Pearson’s Correlation tests against IMPA2 expression. The obtained results of Pearson correlation coefficient (r) for the IMPA2 co-expression in the tested tissues from the two grouped ccRCC patients were then analyzed by GSEA program using Hallmarks gene sets deposited in Molecular Signatures Database (https://www.gsea-msigdb.org/gsea/msigdb)”.
Figure 1 E and F Legends, please explain more.
Response:
As suggested, we have revised the description for Figure 1E as follows. (E and F) Kaplan-Meier analyses for mTORC1 geneset mRNA levels (E) or the signatures of combined IMPA2 and mTORC1 geneset mRNA levels (F) in TCGA ccRCC patients. Others denote the signatures of IMPA2_high/mTORC1_high and IMPA2_low/mTORC1_low. The high and low expression levels of IMPA2 and mTORC1 geneset were determined by the median of their mRNA levels in primary tumors from the TCGA ccRCC database. Please see the revised figure legend for Figure 1E in the revision (lines 223-225).
Line 218, please add the link to the Cancer Cell Line Encyclopedia database
Response:
As requested, we have added the link (https://portals.broadinstitute.org/ccle) to the Cancer Cell Line Encyclopedia database in the revised manuscript (please see line 239 of the revision).
Line 222, there is a typo error, (Fig. 2D) to be changed to (Fig. 2E).
Response:
Thank you so much for your kindly reminding. We have revised this typo error in the revised manuscript (please see line 245 of the revision).
Figure 2 A Legend says (Different letters in each column of b and d indicate the statistical significance at p<0.05) but the bar graph doesn’t show letter d.
Response:
Thank you for your kindly reminding. To avoid misleading, we have changed the labels with symbols to indicate the significant differences in Figure 2A (please see the following figure) and changed “letters” to “symbols” in the figure legend (please see line 252 of the revision).
New Figure 2A:
Figure 2 B Legend needs more explanation.
Response:
As suggested, we have revised the sentence as “Western blot analyses for IMPA2, p-Akt (Tyr308), Akt, p-mTOR (Ser2448), mTOR and GAPDH proteins in whole cell lysates derived from the tested ccRCC cell lines ACHN, Caki-1, Caki-2 and A498”. In the new sentence, we added the Akt and mTOR phosphorylation sites recognized by the antibodies used in the Western blot analyses and the name of the tested ccRCC cell lines (please see lines 254-255 of the revision).
Line 244, please explain more about the autophagy gene set
Response:
As suggested, we have added the link (https://www.gsea-msigdb.org/gsea/msigdb) of Molecular Signatures Database where the audiences are able to find the autophagy gene set used in this study. Also, we revised the sentence as “Accordingly, the transcriptional profiling of IMPA2 and the autophagy gene set obtained from Hallmark gene sets of Molecular Signatures Database (https://www.gsea-msigdb.org/gsea/msigdb) showed a significant (p=0.0049) positive correlation in RCC cell lines from the Cancer Cell Line Encyclopedia database (Fig. 3C)” in the revision (please see lines 269-270).
Figure 4 C & E and Figure 5 C & E legends, please indicate in details which letters indicate the significant differences
Response:
To avoid misleading, we have changed the labels with symbols to indicate the significant differences in Figure 4 C & E and Figure 5 C & E (please see the revised figures as follows) and changed “letters” to “symbols” in the figure legends (please see lines 313, 318, 328, and 334 of the revision).
New Figures 4C and 4E:
New Figures 5C and 5E:
Minor grammatical and spell check required for example,
Line 28 please remove a to read as (Abstract: Although mTOR inhibitors have been approved as first-line therapy for treating)
Line 34 please change to read as (significant inverse correlation in ccRCC tissues).
Line 36 please remove an, to read as (artificially silencing IMPA2 led to increased phosphorylation of Akt/mTORC1 in ccRCC cells).
Line 37 please remove the to read as (The pharmaceutical inhibition of mTORC1 activity by rapamycin reinforced autophagy)
Line 51 please add a before localized disease.
Line 150, please remove by to read as (Statistical analyses were performed using SPSS 17.0 software)
Line 155, remove the extra in
Line 177, please add are instead of is to read as follows (high mTORC1 gene set levels are associated with poor overall survival probability in the TCGA)
Line 194, please change performing to perform
Line 199, please add the before box to read as (The upper and lower lines of the box are the third and first quartiles, respectively).
Line 280, typo error (please change silecncing to silencing).
Line 289, please remove than to read as (tumor colonies over 200 μm in diameter)
Line 290, please add the, to read as (was analyzed by the nonparametric Kruskal Wallis test)
Line 330, please add the, to read as (The upper and lower lines of the box are the third and first quartiles).
Line 332, please add an, to read as (The significant difference was analyzed by an independent sample)
Response:
Thank you so much for kindly reminding us these grammatical errors and typos. We have revised these mistakes (please see lines 28, 34, 36, 37, 51, 164, 169, 191, 211, 216, 307, 316, 317, 361 and 363 of the revision) and carefully checked other grammatical errors and typos throughout the manuscript.
Reviewer 2 Report
This manuscript is well written and results are compelling and clearly described. I have the following questions/comments for the authors:
- How do authors explain the increase of LC3II with higher doses of Rapa in the Western Blot?
- Figure 1, among others, have a legend which feels like an overclaim "IMPA2 downregulation activates...". These figures seem to demonstrate correlation, not causation. Could the authors comment on this?
- In Fig 2D-E and Fig 3D, there are some overclaims. e.g. in Fig 2D the authors claim a decrease in migration ability when they just show changes in protein expression of p-Akt, etc. To support this claim, authors should demonstrate functional changes (e.g., repeat their migration assay after silencing).
Additional minor comments:
- The flow of the introduction could be improved. It is currently jumping from RCC to other maladies (e.g., bipolar disorder), which were not relevant to the conclusions. This distracted from the main point of the paper.
- I would have appreciated more discussion or explanation of the migration experiments. They look difficult to interpret as they stand if you have not performed the technique personally. Assuming that migration graphs are quantifications from the images (e.g., 5C from 5B), I think the manuscript would greatly benefit from more in-depth descriptions of method and result expectations (e.g., more purple signal for more migrated cells).
- line 222- Please, fix the reference to figure 2E, which is currently written as 2D.
- Please, define mTORC1 and autophagosome and draw a relationship between these concepts, your readouts and main story to improve readability.
- The letters used to indicate significances were misleading. I would recommend switching to symbols or roman numbers.
- Please, discuss your results in the discussion briefly.
Author Response
Reviewer 2:
This manuscript is well written and results are compelling and clearly described. I have the following questions/comments for the authors:
Response:
Firstly, we would like to thank reviewer #2 very much for the valuable comments and suggestions to our work.
How do authors explain the increase of LC3II with higher doses of Rapa in the Western Blot?
Response:
Thank you so much for your question. The mechanistic target of rapamycin complex 1 (mTORC1) was unveiled as a master regulator of autophagy since inhibition of mTORC1 was required to initiate the autophagy process. Moreover, LC3 conversion from LC3-I to LC3-II, a phosphatidylethanolamine-conjugated form of LC3-I, is required for the initiation of autophagy assembly. In this study, our data showed that IMPA2 knockdown suppresses the formation of LC3-II (Fig. 4A) and promotes cellular migration ability (Fig.4B). In addition, the reduced expression of IMPA2 correlates with an increased activity of mTORC1 (Fig. 2C) and an enhanced level of phosphorylated mTOR protein (Fig. 2D). Based on these findings, we thought that the increased level of LC3-II detected by Western blot analysis in IMPA2-silenced A498 cells treated with rapamycin should be a consequence of blocking mTORC1 inhibitory function on LC3 conversion.
Figure 1, among others, have a legend which feels like an overclaim "IMPA2 downregulation activates...". These figures seem to demonstrate correlation, not causation. Could the authors comment on this?
Response:
We agree with your points. Indeed, the data shown in Figure 1 merely demonstrate the correlation among the IMPA2 downregulation, an increased activity of the mTORC1-related pathway and the metastatic progression of ccRCC. According to your comments, we have revised the caption of Figure 1 as “IMPA2 downregulation probably correlates with an increased activity of the mTORC1-related pathway and the metastatic progression of ccRCC’ in the revision.
In Fig 2D-E and Fig 3D, there are some overclaims. e.g. in Fig 2D the authors claim a decrease in migration ability when they just show changes in protein expression of p-Akt, etc. To support this claim, authors should demonstrate functional changes (e.g., repeat their migration assay after silencing).
Response:
As suggested, we have revised the sentences for the description for Fig. 2D-E (please see
lines 239-245 of the revision) and Fig. 3D (please see lines 272-273 of the revision) by adding a reference regarding our previous report to illustrate the changes of cellular migration ability after IMPA2 knockdown or overexpression in the tested ccRCC cells. The original and revised sentences as shown in the follows.
(1) Original: Artificially silencing IMPA2 expression promoted cellular migration ability and enhanced the intracellular levels of phosphorylated Akt and mTORC1 proteins in poorly metastatic A498 cells (Fig. 2D). Conversely, the enforced expression of exogenous IMPA2 diminished cellular migration ability and reduced the phosphorylation levels of Akt and mTORC1 proteins in highly metastatic ACHN cells (Fig. 2E).
Revised: Our previous report has demonstrated that IMPA2 knockdown in poorly metastatic A498 cells promotes but IMPA2 overexpression in highly metastatic ACHN cells suppresses cellular migration ability [13]. Here we found that artificially silencing IMPA2 expression enhanced the intracellular levels of phosphorylated Akt and mTORC1 proteins in poorly metastatic A498 cells (Fig. 2D). Conversely, the enforced expression of exogenous IMPA2 reduced the phosphorylation levels of Akt and mTORC1 proteins in highly metastatic ACHN cells (Fig. 2E).
(2) Original: IMPA2 knockdown promoted cellular migration ability and reduced Beclin 1 expression and LC3-II production in poorly metastatic A498 cells (Fig. 3D).
Revised: IMPA2 knockdown reduced Beclin 1 expression and LC3-II production in poorly metastatic A498 cells (Fig. 3D).
Additional minor comments:
The flow of the introduction could be improved. It is currently jumping from RCC to other maladies (e.g., bipolar disorder), which were not relevant to the conclusions. This distracted from the main point of the paper.
Response:
As suggested, we have replaced the sentence ‘Intriguingly, two frequent single-nucleotide polymorphisms (-461C>T and -207T>C) in the promoter sequence of IMPA2, but not IMPA1, of manic-depressive patients support its possible role as a susceptibility gene in bipolar disorder” with “Recent report demonstrated that lithium directly inhibits the enzyme activity of IMPA1, not IMPA2, in the cell-based assays” in order to highlight the differences in the regulation of intracellular IMPA1 and IMPA2 activity. We also added a new Reference 8 (please see lines 61-65 of the revision).
I would have appreciated more discussion or explanation of the migration experiments. They look difficult to interpret as they stand if you have not performed the technique personally.
Assuming that migration graphs are quantifications from the images (e.g., 5C from 5B), I think the manuscript would greatly benefit from more in-depth descriptions of method and result expectations (e.g., more purple signal for more migrated cells).
Response:
Thank you for the constructive suggestion. we have added “purple staining” to indicate more purple signal for more migrated cells in the figure legends (please see lines 250, 311 and 327 of the revision). We have also revised the procedure for the migration experiment in the Materials and Methods in order to making a more detailed explanation (please see lines 119-121 of the revision and the follows).
“In vitro cellular migration ability was assessed by using the Boyden Chamber Assay (NeuroProbe, Gaithersburg, MD, USA). Cells (1.5×104) in serum-free culture medium were added to the upper chamber of the device on a polyvinylidene difluoride (PVDF) membrane with an 8.0 μm pore and precoated with 10 μg /mL fibronectin (Invitrogen) at the opposite site, and the lower chamber was filled with 10% FBS culture medium. After a 3-hour incubation, the membrane was soaked in methanol for 10 min and stained with Giemsa, which was diluted 10-fold by double-distilled water, for 1 hour. Then, the membrane was attached to slides, and the cells on the upper side of the membrane were carefully removed with a cotton swab. The cells on the lower side were photographed. The migrated cells were quantified by counting the cells in three random areas under a microscope at 400x magnification”.
line 222- Please, fix the reference to figure 2E, which is currently written as 2D.
Response:
Thank you so much for your kindly reminding. We have revised this typo error in the revised manuscript (please see line 245 of the revision).
Please, define mTORC1 and autophagosome and draw a relationship between these concepts, your readouts and main story to improve readability.
Response:
As suggested, we have added Fig. 6G in order to draw a relationship between these concepts, our readouts and main story (please see the following figure and figure legend)
(G) Proposed pathway for the IMPA2 downregulation-triggered activation of Akt/mTORC1 and inhibition of autophagy initiation in metastatic ccRCC and a therapeutic value of targeting mTORC1 by its inhibitors, e.g. Rapamycin (Rapa.), to combat metastatic ccRCC with IMPA2 downregulation.
The letters used to indicate significances were misleading. I would recommend switching to symbols or roman numbers.
Response:
As suggested, we have changed the labels with symbols (please see the following figures) in the revision.
New Figure 2A:
New Figures 4C and 4E:
New Figures 5C and 5E:
Please, discuss your results in the discussion briefly.
Response:
As suggested, we have briefly discuss our results in Discussion section of the revision as the following sentences.
“In this study, the in silico experiments demonstrated that IMPA2 downregulation may correlate with an increased activity of mTORC1 in metastatic ccRCC, and this scenario closely associates with a poorer prognosis in ccRCC patients. IMPA2 knockdown in poorly metastatic A498 cells enhanced but IMPA2 overexpression in highly metastatic ACHN cells suppressed the activation of Akt/mTORC1, a negative regulator for autophagy initiation, and the cellular metastatic potentials in vitro and in vivo. The pharmaceutical inhibition of mTORC1 by rapamycin restored autophagy initiation but mitigated the cellular migration ability and lung colony-forming ability of IMPA2-silenced A498 cells. These findings suggest that mTORC1 inhibitors might be useful for treating metastatic ccRCC with IMPA2 downregulation (Fig. 6G)”. Please see lines 385-393 of the revision.
Round 2
Reviewer 2 Report
Thank you for addressing my comments. Almost all of the comments have been addressed in an acceptable manner.
However, one of my comments read "define mTORC1 and autophagosome". While the authors did a good job in illustrating the relationships between them, these terms have not been defined in the introduction, nor anywhere else. e.g. mTORC (mammalian target of rapamycin complex 1 or mechanistic target of rapamycin complex 1). Please, include these changes in your manuscript.
Author Response
Reviewer 2:
Thank you for addressing my comments. Almost all of the comments have been addressed in an acceptable manner.
However, one of my comments read "define mTORC1 and autophagosome". While the authors did a good job in illustrating the relationships between them, these terms have not been defined in the introduction, nor anywhere else. e.g. mTORC (mammalian target of rapamycin complex 1 or mechanistic target of rapamycin complex 1). Please, include these changes in your manuscript.
Response:
Sorry, we did not response to this question in the last revision. As suggested, we have added sentences to define mTORC1 and autophagosome in the Introduction of the revision 2 (please see lines 55-56 and 79-81). The revised sentences are also shown as follows.
Lines 55-56: …………. mechanistic target of rapamycin (mTOR), “is also referred to as the mammalian target of rapamycin and belongs to a member of the phosphatidylinositol 3-kinase (PI3K)-related protein kinases”.
Lines 79-81: A previous report revealed that lithium induces autophagy “which is a spherical structure with double layer membranes and involves in the intracellular degradation system for abnormal proteins or damaged organelles”